# Evaluation of Biochar Nitrate Extraction Methods

**Jenna Walsh, Joseph Sanford * and Rebecca Larson**

Department of Biological Systems Engineering, University of Wisconsin-Madison, Madison, WI 53704, USA
* Correspondence: jrsanford@wisc.edu

**Abstract:** Biochar amendment to soil is a method used to mitigate losses of nitrogen leaching through agricultural soils. Multiple methods for extraction of nitrogen have been used, and recent studies have indicated that traditional soil extraction methods underestimate biochar nitrate. This study evaluated the nitrate extraction efficiency of a KCl extraction method under different temperature (20 and 50 °C) and duration (24 and 96 h) conditions. Increasing the duration of extraction from 24 to 96 h did not have a significant impact on extraction efficiency. However, increasing temperature resulted in nitrate extraction efficiencies above 90%. Rinsing the biochar once with deionized (DI) water following filtration after extraction increased the extraction efficiency significantly, but any subsequent rinses were not significant. This study recommends extracting nitrate from biochar using 2 M KCl at 50 °C for a period of 24 h with one additional rinse to increase nitrate recovery above 90%. However, future studies should evaluate this procedure for different types of biochar produced from alternative biomasses and at varying temperatures.

**Keywords:** biochar; nitrogen; nitrate extraction

## 1. Introduction

Biochar is a carbon-rich substance that is a by-product of slow pyrolysis [1]. Typically, the biochar produced is alkaline, has a high surface area, and has a large cation exchange capacity [2,3]. These properties have made it attractive for various end uses, including nitrogen management. Nitrogen fertilizers are essential to crop production, but application often results in a variety of losses. Production of nitrous oxide—a greenhouse gas—can occur under certain soil conditions, and nitrogen-rich wastewater can degrade surface and ground water leading to serious impacts on environmental and human health [4]. Improving soil nitrogen retention is an important strategy to mitigate these risks as well as increase crop yields. Nitrates however, can be particularly challenging to manage as they are highly mobile in soil systems [5]. Nitrates are known to cause health issues when consumed at certain concentrations, so limiting their movement to drinking water sources is especially important.

Biochar has shown potential to accumulate nitrogen in many forms [6]. A number of studies have found reductions in nitrate leaching from biochar amended soils [7–9]. It has been hypothesized that certain conditions can lead to nitrate retention, however the mechanisms are currently unknown [10–13]. In order to better assess and improve the performance of these systems, sound methodologies are needed to quantify nitrate uptake by biochar, which requires nitrate extraction for measurement. The most common method used to extract nitrate from biochar is derived from the soil KCl inorganic nitrogen extraction method [14]. This method involves mixing soil or biochar 1:4 (w/v) with a 2 M KCl solution and agitating for one hour at room temperature. However, it has been shown that this method results in incomplete nitrate recovery from biochar [12,13,15,16]. Several studies have shown that repeated KCl extraction increased the mass of nitrate extracted from cocomposted biochar over the standard one-hour extraction, with higher temperatures and longer agitation further increasing

extraction [12,13,15]. However, the initial amount of nitrate held by the biochar in these studies was unknown, so extraction efficiency cannot be determined. In this study, nitrate extraction methods are examined to determine the procedures that maximize extraction efficiency and quantify the mass fraction extracted.

## 2. Materials and Methods

Nitrate extraction efficiency was measured using biochar produced by Carbon Terra from wood chips (80% coniferous, 20% deciduous) at a pyrolysis temperature of 700 °C for 36 h. While biochar can be produced from a variety of biomasses at varying temperatures, Carbon Terra biochar was chosen for analysis as it has been used in a previous study where nitrate extraction methods were identified as having low nitrate recovery after composting [11]. Prior to oxidation, nitrate sorption, and extraction, biochar composition of C, H, and N was determined to be 86.0%, 4.6%, and 0.2%, respectively, using a PerkinElmer Elemental Analyzer (PerkinElmer, Waltham, MA, USA), following method ASTM D537. Oxygen content was determined to be 7.6%, based on the remaining composition after subtracting C, H, N, ash (via ASTM D1102 using a muffle furnace at 600 °C for 2 h). The pH was measured using a HACH HQ440d Benchtop Multi Meter pH probe (Hach Company, Loveland, CO, USA), after diluting the biochar 1:20 by mass with DI water, and was determined to be 9.4 for the initial biochar. In this study, biochar with a particle size of 2.00 to 4.75 mm was used.

As biochar nitrate sorption is low without modification or aging, biochar was oxidized using a sodium hypochlorite ($NaClO$) oxidation method [10]. All oxidized biochar was dried at 105 °C for at least 24 h before beginning the nitrate sorption process. A 50 mg $NO_3$-N $L^{-1}$ stock solution made from sodium nitrate ($NaNO_3$) was mixed with 2.5 g oxidized biochar at a ratio of 1:10 (w/v). Control samples containing only stock solution were included. Samples were covered and shaken at 100 rpm on a G10 Gyrotory 141 Shaker (New Brunswick Scientific Co., Edison, NJ, USA) for 24 h at room temperature. Samples were then vacuum filtered through 1.5 μm filters, and the filtrate was acidified with $H_2SO_4$ and stored at 4 °C until analysis. Biochar was transferred to clean bottles and stored at 4 °C until extraction. All nitrate solutions were analyzed using a SEAL AQ2 automated discrete analyzer (SEAL Analytical Inc., Werkstrasse, Germany) following the EPA-126-A Rev 9 [17]. Nitrate absorbed was determined by the nitrate concentration difference in the initial and final solutions.

Following nitrate sorption, the biochar was extracted using methods that varied in extraction duration and temperature, Table 1. All methods reported in this paper used a KCl extraction solution. Different extraction solutions were initially evaluated (0.5 M $K_2SO_4$ and 0.1 M HCl) but were excluded from this paper due to recovery efficiencies below 20%. There are several nitrate extractants that could be used (i.e., $CaSO_4$, $NH_4F$, $H_2SO_4$, $CaCl_2$, $NaHCO_3$, $CuSO_4$, $Ag_2SO_4$), nonetheless, 2 M KCl was chosen because it is the most commonly used and accepted method for determining inorganic nitrogen [14,15]. In this study, biochar was mixed with 25 mL of a 2 M KCl solution to an approximate ratio of 1:10 w/v and shaken at room temperature for 24 h (RT24), longer than the one-hour in the above methods, as suggested by Kammann et al. [11]. After an initial vacuum filtration, three rinses were performed by pouring 25 mL of deionized (DI) water over biochar and vacuum filtering. Filtrate from each rinse was acidified with $H_2SO_4$ and stored at 4 °C until analysis. Biochar was dried in an oven at 105 °C for 24 h and then weighed to account for any biochar losses during the extraction process. Extraction duration and temperature were evaluated. Past studies have found that standard extraction durations are not sufficient to remove nitrate form biochar, thus, using a 24 h extraction has been suggested [11]. However, past studies could not determine the efficiency due to biochar being from field or cocompost studies, thus, it is not established if going beyond 24 h would result in higher extraction efficiency. Therefore, shaking time was increased to 96 h to evaluate if increasing the time by four times, as in previously suggested methods, would have an impact. Additionally, the influence of temperature was evaluated. Hadier et al. [15] previously suggested that temperature of extraction should be increased to 50 °C because when temperatures reaches 50 °C extractant can penetrate into the 3-dimensional inner pores [18], which could increase efficiency of extraction and

shorten the duration needed to achieve equilibrium. Overall, four different extraction treatments were evaluated, with modifications to extraction duration and temperature, Table 1, including room temperature for 24 h (RT24), room temperature for 96 h (RT96), 50 °C for 24 h (HT24), and 50 °C for 96 h (HT96). All extractions were conducted in triplicates and control samples containing biochar without the nitrate solution (in DI water) were included to measure any nitrate lost from the raw biochar.

**Table 1.** Biochar extraction methods.

| Treatment | Temperature (°C) | Duration (h) |
|-----------|------------------|--------------|
| RT24 | 20 | 24 |
| RT96 | 20 | 96 |
| HT24 | 50 | 24 |
| HT96 | 50 | 96 |

Statistical analysis was performed using SAS version 9.4 (SAS Institute, Cary, NC, USA) [19]. A one-way ANOVA and Tukey HSD were performed to assess the differences between extraction methods and rinses. Additionally, two-way ANOVA was used to compare the effects of temperature and duration on nitrate extraction. All analyses used a *p*-value of 0.05.

## 3. Results

Recovery efficiencies ranged from 69%–87% for the initial extraction, Figure 1. The first rinse significantly increased the recovery for all treatments with the exception of RT96 (that treatment had high variability). However, each subsequent rinse (rinse two and three) resulted in a smaller recovery of nitrate which did not significantly increase nitrate extraction efficiency. Comparing treatments, temperature was the primary driver for extraction efficiency as the increased temperature from 20 to 50 °C resulted in a significant increase in nitrate extraction efficiency, Figure 1. This is likely due to temperature influencing the ability of extractants to penetrate the pore space in biochar. Increased extraction temperatures result in weaker water biochar surface interactions [17], allowing extractants to penetrate smaller biochar pores to extract nitrate [15].

Treatment duration did not significantly impact nitrate extraction efficiency following the first rinse, Figure 1. It was expected that allowing longer exposure time to KCl would result in higher extraction efficiency, similar to what was observed when duration was increased from one to 24 h in previous studies [15]. However, increasing the time to 96 h did not significantly impact extraction efficiency, thus, 24 h is likely suitable for extraction. However, it may be possible to achieve an appropriate extraction efficiency at shorter durations, and future studies should evaluate shaking times between one and 24 h to identify if procedure time could be reduced.

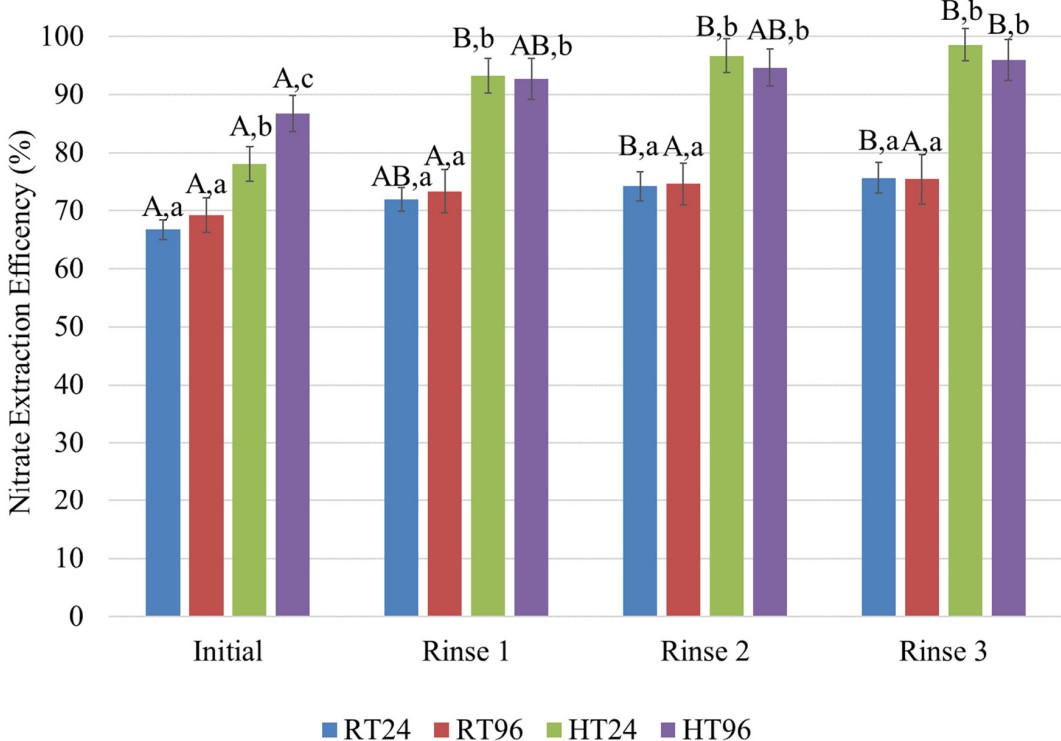

**Figure 1.** Nitrate extraction efficiency from biochar following initial KCl extraction and subsequent rinses. Error bars represent standard deviation between replicates. Capital letters signify a significant difference between nitrate extraction efficiencies for initial and subsequent rinses within each treatment; lowercase letters signify a significant difference between treatments for the extraction step among treatments.

## 4. Conclusions

Using traditional nitrate extraction methods for biochar may underestimate the nitrate content of the biochar. Increasing the temperature of the extraction to 50 °C for a period of 24 h results in an extraction efficiency above 90% after one additional rinse. Additional rinses beyond the first rinse did increase the extraction efficiency, although the fraction removed after each rinse was not significant. It is recommended that, to increase nitrate extraction efficiency for biochar, the temperature and duration of extraction should be increased from other methods, particularly those developed for soil matrices. In this case, it is recommended to use 50 °C for a period of 24 h with one additional rinse to increase nitrate recovery above 90%. However, future studies should investigate if shorter durations would achieve similar results, as there was no significant difference between 24 and 96 h in this study and it may be possible to achieve equilibrium from shorter durations. Additionally, this study was limited in scope to only one biochar and one extractant, thus, future studies should investigate extraction efficiency for biochars produced from different biomasses, temperatures, and modifications to confirm the method's performance, and evaluate other extractant possibilities to see if the current method could be altered for better performance.

**Author Contributions:** The authors contributed equally to this work.

**Funding:** This material is based upon work that is supported by the National Institute of Food and Agriculture, U.S. Department of Agriculture under project numbers 2015-67019-23573 and 2017-67003-26055. Any opinions, findings, conclusions, or recommendations expressed in this publication are those of the author(s) and do not necessarily reflect the view of the U.S. Department of Agriculture.

**Acknowledgments:** The authors would like to acknowledge Hui Wang, Esmeralda Tovar, Victoria Shveytser, and Daniel Coleman for laboratory assistance.

**Conflicts of Interest:** The authors declare no conflict of interest.

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
