# Peer review of "Evaluation of Biochar Nitrate Extraction Methods"

_applsci, doi:10.3390/app9173514_

Round 1
Reviewer 1 Report
Your research is well conducted!!
Author Response
Thank you for taking the time to review our manuscript. We are glad you thought the research was well conducted and appreciate your time.
Reviewer 2 Report
Applied Sciences (ISSN 2076-3417)
Manuscript No.: applsci-559183
Title: Evaluation of Biochar Nitrate Extraction Methods
Manuscript debates the nitrate extraction methods of biochar which is really needed for a possible using of biochar in agriculture as a mineral fertilizer.
Major comments:
Even if the manuscript shows some interesting data the present manuscript brings not such a novelty. E.g. DOI: 10.1016/j.geoderma.2018.11.006 (reference is not mine) is using CaCl2 extraction method. So it would be nice to compare CaCl2 and KCl solutions of biochar nitrate extraction methods. Furthermore, it's a pity that authors did not use some other biochar of different type of biomass and temperature (range 250-700°C). Why the only one type of biochar was examined? How the results will be affected if one will use e.g. biochar of wheat straw prepared at 300°C? Before a possible publication I suggest to test a three type biochar in relation to different temperature (e.g. 250, 500 and 700°C). Compare the results between each other and briefly discuss with literature when using other extraction solutions. It would be also nice to see how the efficiencies will change in biochar amended soils.
Minor comments:
L32: "Sorption" or "accumulation" could be more appropriate instead of "uptake".
L85: Treatments abbreviations should be explained first.
L96: Add some statistical evaluation among the treatments (RT24…HT96).
L97: Rewrite "rises" to "rinses".
L104: If you add letters that represent significant difference between nitrate extraction efficiencies for initial and subsequent rinses into the Figure 1 the Table 2 is redundant.
Reviewer 3 Report
This is a well written and well - presented manuscript. The rationale for the study is sound and the research aims are clearly stated. The style is succinct for a communications manuscript. The recommendations in the conclusions are also clearly stated and are a direct interpretation of the results.
I have two specific questions.
Biochar can be produced at lower temperatures. Can the authors insert some text to note this and also explain why only Carbon Terra biochar was used? An expansion of the experimental investigation could have been to compare extraction efficiencies using biochar produced at different temperatures and possibly different oxidation regimes.Recommendation: I would like to see some text to explain the rationale for only using Carbon Terra biochar (and noting that biochar can be produced at other temperatures).
Table 1. Why were only 2 temperatures and 2 durations used? While the authors mention that different extraction solutions were used but resulted in low recovery efficiencies (L72-73), an expanded experimental design could have identified clearly when a maximum recovery was effected (i.e. testing longer and shorter extraction durations; and longer and sorter temperatures). Perhaps this has been done (L72-73?) or as cited in the literature (14, 15).Recommendation: some text noting if more temperatures and duration of extraction were used, or if why only 2 of each were used (ie literature recommendations?), would help to bring some context to the methods reported here.
I realise this is a communication manuscript, so just a sentence or two for both points would be sufficient to add the necessary context for my queries.
Round 2
Reviewer 2 Report
Authors really enhanced the previous version of the manuscript. I agree with all of the changes. I have no more questions or recommendations. I can suggest the present version of manuscript to publish as a short communication in Applied Sciences journal.